# The Role of *Aspergillus niger* in Regulating Internal Browning Involves Flavonoid Biosynthesis and the Endophytic Fungal Community of Pineapple

**DOI:** 10.3390/jof10110794

**Published:** 2024-11-15

**Authors:** Fei Shen, Guang Wang, Shijiang Zhu

**Affiliations:** Guangdong Province Key Laboratory of Postharvest Physiology and Technology of Fruit and Vegetables, College of Horticulture, South China Agricultural University, Guangzhou 510642, China; 18819266981@139.com (F.S.); wguang@scau.edu.cn (G.W.)

**Keywords:** pineapple, internal browning, endophytic fungus, *Aspergillus niger*, flavonoid

## Abstract

Endophytic fungi are commonly used to control plant diseases, overcoming the drawbacks of chemical agents. The internal browning (IB) of postharvest pineapple fruit, a physiological disease, leads to quality losses and limits industrial development. This work investigated the relationship among the effects of *Aspergillus niger* (An) on IB controlling, flavonoid metabolism and the endophytic fungal community of pineapple through metabolomics, transcriptomics, microbiomics and microorganism mutagenesis technology. We obtained an endophyte An that can control the IB of pineapple and screened its mutant strain AnM, through chemical mutagenesis, that cannot control IB. The transcriptome of fungi showed that An and AnM were different in oxidative metabolism. Transcriptome and metabolome analyses of pineapple showed that An upregulated genes of flavonoid synthesis, including *dihydroflavonol 4-reductase* and *flavonoid 3′-monooxygenase* and increased the flavonoid content in pineapple fruit, i.e., Hispidulin, Hispidulin-7-O-Glucoside, and Diosmetin, while AnM could not. Microbiomics analysis identified an increase in the abundance of eight endophytic fungi in An-inoculated fruit, among which the abundance of six endophytic fungi (*Filobasidium magnum*, *Naganishia albida*, *A. niger*, *Aureobasidium melanogenum*, *Kwoniella heveanensis* and *Lysurus cruciatus*) was positively correlated with the content of three flavonoids mentioned above but not in AnM-inoculated fruit. Overall, this suggested, for the first time, that *A. niger* alleviated IB mainly by enhancing flavonoid synthesis and content and the abundance of endophytic fungi and by regulating the interaction between flavonoid content and endophytic fungi abundance in pineapple. This work adds to the understanding of the IB mechanism in postharvest pineapple and provides a new green approach for reducing postharvest losses and controlling physiological diseases.

## 1. Introduction

Pineapple dominated the world trade of tropical fruits, and the world production of pineapple increased at a different rate per year during the past decade, according to Food and Agriculture Organization of the United Nations (FAO) statistics in 2020. But pineapple is susceptible to internal browning (IB) during the postharvest storage. IB, a physiological disorder of pineapple, seriously impacts the profitability of the pineapple industry [1].

Current methods for controlling the IB of postharvest pineapple fruit include cultivating resistant varieties [2,3,4], applying chemical treatments, such as ascorbic acid, silicon, calcium salts and plant hormones [5,6,7,8,9] and using physical techniques, like waxing, modified atmosphere packaging, hot air treatment and cold storage [10,11,12,13,14]. For example, postharvest treatment with a calcium salt solution could delay the oxidation of cell membranes and reduce the activity of polyphenol oxidase (PPO) in pineapples, thus helping to control IB [9]. Spraying pineapples with abscisic acid could control IB during storage at ambient temperature by inhibiting the synthesis and oxidation of phenolic compounds, enhancing the antioxidant capacity of fruit and regulating the balance of endogenous hormones [15,16]. Song et al. [13] indicated that hot air treatment alleviated IB by activating the oxidative stress system and regulating oxidative stress in pineapples. Shen et al. [17] reported, for the first time, that exogenous inoculation with the endophytic fungus *Penicillium* sp. could suppress the development of IB in pineapple. This study remains the only one to date that addresses the control of physiological disorders in plants using an endophytic fungus. In recent years, RNA-sequencing technology and metabolomics analysis have been employed to clarify the mechanisms of IB in postharvest pineapples, including studies on gene expression profiles in response to chemical treatments and physical methods [5,13]. However, research on the effects and mechanisms of endophytic fungi in controlling IB in pineapple fruit remains limited.

Flavonoids, a diverse group of secondary metabolites found in plants, have attracted growing attention for their potential applications in plant disease management. Research has demonstrated that flavonoids and their biological activities can help control plant diseases by defending against pathogens [18,19,20,21]. Flavonoids exhibit their antifungal and antibacterial properties, inhibiting pathogen growth and modulating plant immune responses [22,23,24,25,26,27,28,29,30]. For example, naringenin has been reported to inhibit anthracnose of poplar caused by *Collelotrichum gloeosporioides* [23], while the isoflavone genistein enhanced soybean resistance to damage caused by *X*. *axonopodis* pv. *Glycines* [27]. The resistance of rice to rice blast has been linked to isoflavone synthesis in the plant [28]. Flavonoids are also known to modulate the stress resistance of plants [31,32,33]. In maize, *C*-glycosyl flavonoids serve as biomarkers for the plant’s response to cold stress in the field [32]. However, the relationship between flavonoid regulation and the control of IB in pineapples remains unclear.

Endophytic microorganisms, including bacteria, fungi and archaea, live within plant tissues and form symbiotic interactions with their host plants [34,35,36,37]. These endophytes have a vital impact on the production and regulation of secondary metabolites, which are essential for plant growth, development, defense mechanisms, stress responses and adaptation to environmental changes [38,39,40,41]. Endophytic microorganisms have been reported to enhance the biosynthesis of secondary metabolites by activating specific pathways within plants [38,39,42]. For instance, the endophytic fungus *Piriformospora indica*, found in roots, has been shown to promote growth, nutrient uptake and abiotic stress tolerance in soybean (*Glycine max*) by stimulating the biosynthesis of flavonoids and flavonols through the phenylpropanoid and derivative pathways [43]. Endophytes play a key role in modulating plant defense mechanisms, often by producing secondary metabolites that deter herbivores or insects [41,44,45,46] and provide protection against pathogens [35,37,41]. Studies reported that root-associated mutualists stimulated the induced systemic resistance (ISR) of plants by regulating phytohormones and proteins of defense in aboveground plant parts to protect plants from pathogen attacks and improve plant health [47,48,49]. In endophyte-infected tall fescue, silicon accumulation [50] and alkaloid production [51] have been shown to reduce herbivory by generating compounds that are toxic or unpalatable to insects and grazing animals. However, no studies to date have explored the role of endophytes in controlling plant physiological disorders through the involvement of secondary metabolites.

Microbial mutagenesis breeding is a widely used breeding method that can shorten the microbial breeding cycle, improve breeding efficiency and produce strains with desired traits [52,53,54]. In recent years, there has been growing interest in applying the mutant strains obtained through microbial mutagenesis breeding for the control of plant diseases [55,56,57,58,59,60]. In vivo experiments showed that the mutation of strains of *Kloechera apiculata*, induced by ultraviolet and microwave irradiation, significantly reduced the incidence of blue mold (*Penicillium italium* Wehmer) and green mold (*P. digitatum* Sacc.) in postharvest citrus compared to the starting strain [61,62]. Additionally, the antagonistic activity of a mutant strain of *Bacillus subtilis,* obtained through mutagenesis breeding, was found to be higher than that of the wild-type stain, making it effective for controlling root rot in soybean, watermelon wilt and clubroot in Cruciferae [56,59,63,64]. However, no research has explored the mechanisms by which mutant strains of endophytes control endophyte control of plant physiological disorders through mutagenesis breeding.

The enhancement of antioxidant ability in pineapple is closely linked to the inhibition of IB in postharvest fruit [9]. Flavonoids, known for their antioxidant properties [29,65], have been shown to positively correlate with the antioxidant activity of plants [30,66]. Endophytes can control plant diseases by regulating the microbial community structure in plants [67,68,69]. However, the relationship between flavonoid content in pineapples and IB has not been studied, nor is it known whether IB is related to changes in the microbial community structure in pineapples. In this study, the endophytic fungus *Aspergillus niger* and its mutant strains, obtained through mutagenesis breeding, were inoculated with pineapple to compare their effects on controlling IB. Through the methods of metabolomics, transcriptomics and microbiomics, this study explores, for the first time, the effects of flavonoids and changes in the microbial community structure mediated by the exogenous endophytic fungus *A. niger* on pineapple IB. The goal is to provide a theoretical foundation for uncovering the mechanism by which endophytes prevent IB in pineapple.

## 2. Methods

### 2.1. Acquisition and Cultivation of Endophytic Fungus and the Mutant Strain

The endophytic fungus *Aspergillus niger* isolated from pineapple is preserved in the Guangdong Province Key Laboratory of Postharvest Physiology and Technology of Fruits and Vegetables, College of Horticulture, South China Agricultural University, Guangzhou, Guangdong Province.

The mutant strain AnM was obtained through the chemical mutagenesis of *A. niger* spores using diethyl sulfate solution, according to the method of Song et al. [70]. Spores of *A. niger* (final concentration of 10^6^ cfu·mL^−1^) and diethyl sulfate solution (20% *v*/*v*, the final concentration of 0.5%) were mixed and homogenized and then shook for 30 min (100 r·min^−1^). The sodium thiosulfate (150 μL, 25% *w*/*v*) was added into the solution mixture to end the mutagenic reaction. Finally, the treated spore suspensions were inoculated onto potato dextrose agar (PDA) medium and cultured (28 °C, at the darkness) to obtain the mutant strains.

*A. niger* and its mutant strain AnM were cultured aerobically on PDA medium at 28 °C in darkness for 16 days, after which they were used for experiments involving pineapple fruit.

### 2.2. Plant Materials and Treatment

Pineapple (*Ananas comosus* L. cv ‘Comte de Paris’) fruit at uniform maturity of 70% (approximately 120 days after flowering) [15] and a range of approximately 500 g, without disease or insect injury and physical damage, were picked from a business agricultural farm in Xuwen County, Guangdong Province, China. Pineapple fruit were cleaned in running water and then dried in the air before treatments were performed. In our study, two different experiments were conducted. The IB rate was evaluated after a 12 d storage.

Experiment (A): pineapple fruit were randomly distributed among three groups, containing 20 fruit per group. Then, pineapple fruit were packaged into polyethylene films (50 µm) and stored at 20 ± 1 °C and a relative humidity (RH) of 90–95% in darkness. Samples of pulp tissues were collected as just harvested (stored 0 d, ST.0d) and stored for 12 days (ST.12d), quick-frozen using liquid nitrogen and then stored at −80 °C until use for the detection of flavonoid levels. The aim of this experiment was to research the relationship between the rate of IB and flavonoid content in pineapple during the storage.

Experiment (B): there were three treatments, with pineapple fruit being spray-inoculated with sterile water (CK), the spore suspension (5 × 10^7^ cfu·mL^−1^, the spore suspension was made in sterile water) of the endophytic fungus *A. niger* (An) and the mutant strain AnM (AnM). Each treatment was performed with three replications, 20 fruit per replication. After treatment, pineapples were packaged and stored the same as in Experiment (A). Samples of pulp tissues of pineapple fruit were collected at 12 days (12 d), quick-frozen with liquid nitrogen and then stored at −80 °C until use for the detection of flavonoid levels and analysis of the transcriptome and endophytic fungal community structure in pineapple fruit. The pineapple pulp samples were obtained as follows: after disinfecting the surface of the pineapple fruit twice with a 75% ethanol solution, the fruit was placed on a clean bench, the pericarp, core and crown of the fruit were removed and the pulp was collected (all tools are sterile). This is performed to check whether *A. niger* alleviation of pineapple internal browning (IB) involves the flavonoid content (metabolome), transcriptome and endophytic fungal community structure of pineapple fruit.

### 2.3. Internal Browning Severity Assessment

The assessment of IB incidence was based on the previous approaches [15].

### 2.4. Extraction and HPLC-ESI-MS/MS (Metabolome) Analysis of Flavonoids

Metabolome analysis of flavonoids referred to the previous approaches [6]. The multiple reaction monitoring (MRM) was managed using Metwar Biotechnology Co., Ltd. (Wuhan, China). Total flavonoids were extracted from the freeze-dried pineapple pulp samples (100 mg) using a methanol/water (70:30, *v*/*v*, 1.2 mL) solution at 4 °C. The extracts were centrifuged at 4 °C (12,000× *g*, 10 min). The supernatants were gained and filtered via filtration (pore size: 0.22 μm; Anpel, Shanghai, China). Then, the filtrated liquid was determinated according to the B4500 Q TRAP UPLC/MS/MS platform (UPLC, SHIMADZU Nexera X2; MS, Applied Biosystems 4500 Triple Quadrupole) (Shimadzu Corporation, Kyoto, Japan). Separation of the filtrates was reached at 40 °C and the flow rate of 0.35 mL·min^−1^ on a SB-C18 column (1.8 µm, 2.1 mm × 100 mm, Agilent, Santa Clara, CA, USA) and the injection volume was 4 μL. Elution was executed with mobile phase A (0.1% formic acid solution of ultrapure water) and mobile phase B (acetonitrile solution of 0.1% formic acid). The gradient procedure was 95:5 *v*/*v* at 0 min, 5:95 *v*/*v* at 9 min, hold for 1 min, 95:5 *v*/*v* at 10–11.10 min and keep for 5 min. An ESI-triple quadrupole-linear ion trap (Q TRAP)-MS was alternatively linked to the effluent. Using a triple quadrupole-linear ion trap mass spectrometer (Q TRAP), LIT and triple quadrupole (QQQ) scans were achieved, with the AB4500 Q TRAP UPLC/MS/MS System, outfitted with an ESI Turbo Ion-Spray interface (Agilent Technologies, Santa Clara, CA, USA), performing it in a positive ion manner and managed with software of Analyst 1.6.3 (AB Sciex, Toronto, CAN). The performance attributes of the ESI source were as follows: ion source, turbo pray; source temperature, 550 °C; ion spray voltage (IS), 5500 V (positive ion manner); ion source gas I, 50 psi; ion source gas II, 60 psi; curtain gas (CUR), 35 psi. flavonoids were regarded as significantly differentially accumulated with a VIP ≥ 1 and fold-change ≥ 2.0 (up-regulation of flavonoid content) or fold-change ≤ 0.5 (down-regulation of flavonoid content).

### 2.5. Transcriptome Analysis of Fungi

Total RNA was obtained from the hyphae and spores of *A. niger* and AnM samples with the RNAprep Pure Plant of Polysaccharides and Polyphenolics-rich Plus Kit (DP441, Tiangen, Beijing, China), based on the manufacturer’s description. The total contents and integrity of RNA were evaluated using the RNA Nano 6000Assay Kit of the Bioanalyzer 2100 system (Agilent Technologies, Santa Clara, CA, USA). cDNA fragments (length of 370–420 bp) were selected, using the AMPure XP system (Beckman Coulter, Beverly, MA, USA), to purify the library fragments. Then, PCR amplification was performed, using AMPure XP beads (Beckman Coulter, Beverly, MA, USA), to cleanse the PCR product and finally achieve the library. With Qubit2.0 Fluorometer (Invitrogen, Santa Clara, CA, USA), the library was tested and measured and then sequenced using the Illumina NovaSeq 6000 (Illumina, San Diego, CA, USA). To guarantee the quality and reliability of sequencing data analysis, the raw data were screened to obtain clean data. The software Trinity (version 2.6.6) was implemented to assemble the clean data. The software of BUSCO (Benchmarking Universal Single-Copy Orthologs) (version 5.0) was used to assess the assembling quality of Trinity. Gene functional annotation was performed with databases of Nr (NCBI non-redundant protein sequences), Nt (NCBI non-redundant nucleotide sequences), Pfam (Protein family), KOG/COG (Clusters of Orthologous Groups of proteins), Swiss-Prot (A manually annotated and reviewed protein sequence database), KO (KEGG Ort holog database) and GO (Gene Ontology). The DESeq2 R package (version 1.20.0) was implemented to analyze differential expression analysis of the two treatments. The threshold of *padj* < 0.05 and |log2(fold-change)| > 0.5 were determined to be significantly differential expression. GO function enrichment analysis and KEGG pathway enrichment analysis of differential gene sets were executed with the software GOseq (version 1.10.0) and KOBAS (version 2.0.12).

### 2.6. Transcriptome Analysis of Pineapple Fruit

The RNAprep Pure Plant Plus Kit (DP441, Tiangen, Beijing, China) was implemented to extract total RNA from the samples of pineapple pulp. The NanoPhotometer^®^ spectrophotometer (IMPLEN, Los Angeles, CA, USA) was used to quantify RNA, and the RNA Nano 6000 Assay Kit of the Bioanalyzer 2100 system (Agilent Technologies, Santa Clara, CA, USA) was employed to evaluate the integrity of the RNA. Sequencing libraries were prepared using the NEBNext^®^ Ultra^TM^ RNA Library Prep Kit for Illumina^®^ (NEB, Ipswich, MA, USA) according to the manufacturer’s approaches. The AMPure XP system (Beckman Coulter, Beverly, MA, USA) was implemented to designate fragments of 250–300 bp in length, which were then PCR-amplified. The AMPure XP system was employed to depurate the PCR products. Using the Agilent Bioanalyzer 2100 system, library quality was evaluated. Finally, an Illumina Novaseq platform was used to sequence six cDNA libraries (the Novegene Technology Company, Beijing, China).

Raw reads (raw data) were performed by in-house perl scripts to rid low quality reads and reads including adapter and *n* to gain clean reads firstly. Then, the clean reads (clean data) were mapped to the pineapple reference genome [71] with Hisat2 (version 2.0.5), and the prediction of novel genes was managed with String Tie (version 1.3.3b) [72]. The analysis of RNA differential expression was calculated using DESeq2 software (version 1.20.0) between the two different treatments. The genes with the threshold of *Padj* ≤ 0.05 and |Log2 (fold-change)| ≥ 1 were considered as significant differential expression genes (DEGs). Gene ontology (GO) enrichment analysis and Kyoto Encyclopedia of Genes and Genomes (KEGG) pathway enrichment were used to annotating the pathways representative of DEGs.

### 2.7. Analysis of Endophytic Fungal Community of Pineapple Fruit

The analysis of the endophytic fungal community was performed via ITS rRNA amplicon sequencing (the Novegene Technology Company, Beijing, China). The total DNA from the samples of pineapple pulp were extracted using the CTAB protocol [73,74]. The primer for fungi was ITS5-1737F and ITS2-2043R [75]. The PCR was conducted with initial denaturation at 98 °C for 1 min, subsequent to 30 cycles of denaturation at 98 °C for 10 s, annealing at 50 °C for 30 s, elongation at 72 °C for 30 s and a final extension at 72 °C for 5 min. Then, the Qiagen Gel Extraction Kit (Qiagen, Hilden, Germany) was engaged to further depurate the PCR products. The TruSeq^®^ DNA PCR-Free Sample Preparation Kit (Illumina, San Diego, CA, USA) was engaged to generate sequencing libraries according to the manufacturer’s instructions, and index codes were added. The Qubit@ 2.0 Fluorometer (Thermo Scientific, Waltham, MA, USA) and system of the Agilent Bioanalyzer 2100 (Agilent Technologies, Santa Clara, CA, USA) were applied to analyze the library quality. At last, an Illumina NovaSeq platform was utilized to sequence the library, resulting in the generation of 250 bp paired-end reads, which were formed. Sequence analysis was managed using Uparse software (version 7.0.1001) [76], and the Mothur algorithm was employed to annotate taxonomic data based on the Silva Database [77]. Alpha diversity and *beta* diversity indices were carried out with QIIME (version 1.7.0 and version 1.9.0, respectively). To predict the functional profiles of the endophytic fungi community, FunGuild [78] was performed, estimating the functions based on ITS rRNA sequencing data.

### 2.8. Statistical Analysis

As per experiments, three independent biological replicates were executed. The software of Origin2020 was employed for statistical analysis. One way or multi-way analysis of variance (ANOVA) was implemented to estimate the data. The least significant difference (LSD) was estimated and significance was conducted at the level of *p* < 0.05.

## 3. Results

### 3.1. Changes in IB Incidence and Flavonoid Content in Pineapple Fruit During Storage

In order to investigate the relationship between IB incidence and flavonoid content in pineapple, IB incidence and flavonoid content in pineapple pulp were analyzed before and after storage. As shown in Figure 1A,B, the IB incidence of ST.12d (storage at 12 days) pineapple was significantly higher than that those just harvested (stored 0 d, ST.0d). To clarify the relationship of pineapple IB between flavonoids, UPLC-MS/MS was applied with samples with obvious IB (ST.12d) compared with those with no IB (ST.0d). Results showed that 20 flavonoids in pineapple pulp were identified. Of all the flavonoid components, only one was significantly increased following 12 d of storage (i.e., Catechin), while the others decreased significantly. These results imply that IB of pineapple aggravated and the flavonoid contents in pineapple, which were decreased with the extension of the storage time.

### 3.2. RNA Sequencing, De Novo Assembly and Gene Annotation of Fungus Transcriptome

In order to check whether there were differences in transcription levels between *A. niger* and its mutant strain AnM, transcriptome profiles of the two strains were analyzed. For this objective, the cDNA library established from the total RNA of strains was analyzed using the Illumina Novaseq platform. Following removal of the adapters, ambiguous nucleotides and low-quality reads, the sum of 37.58 gigabytes (Gb) of clean bases, including 250,527,830 clean reads and 11,788 genes, was achieved (Table 1 and Appendix A). The average clear reads per sample was approximately 6.26 Gb with a GC percentage of 54.24% to 54.99% (Table 1). The Q20 (the percentage of bases with Phred > 20 to the total bases, Phrede = −10log10 (e)) and Q30 (the percentage of bases with Phred > 30 to the total bases, Phrede = −10log10 (e)) were both approximately 97.6%. Through alignment with the splicing sequence of Trinity, 88.55-90.62% of the reads in the nine libraries were uniquely mapped (Table 1). The BUSCO software was used to evaluate cluster.fasta, trinity.fasta and unigene.fasta obtained using splicing reads. As shown in Appendix A, the complete single-copy BUSCOs and complete duplicated BUSCOs (C) were 269, 286 and 268, respectively, accounting for a relatively high proportion of the total BUSCOs, as follows: 92.76%, 98.62% and 92.41% (Appendix A); the percentages of missing BUSCOs of cluster.fasta, trinity.fasta and unigene.fasta were 1.34%, 0.69% and 1.03%, indicating that the integrity and quality of the splicing sequence were good. All the above results prove that the veracity and quality of the sequencing reads achieved the standards for deeper analysis. Principal component analysis (PCA) revealed that the endophyte *A. niger* and its mutant strain AnM were in two significantly distinct groups (Appendix A), indicating that the transcription level of the endophytic fungus *A. niger* (An) was different from that of the mutant strain AnM.

### 3.3. Transcriptomic Changes in Endophyte A. Niger and Its Mutant Strain AnM

There were 7548 co-expressed genes of the endophytic fungus *A. niger* and its mutant strain AnM, 368 specific genes of *A. niger* and 1640 of AnM (Figure 2A). As shown in Figure 2B, a total of 2391 DEGs were discovered in AnMvsAn (the endophytic fungus *A. niger* compared to the mutant strain AnM), including 1279 up-regulated DEGs and 1112 down-regulated. These results prove that there were significant differences in transcription levels between *A. niger* and its mutant strain AnM. Figure 2C shows the 20 most enriched GO terms identified in the fungus (AnMvsAn). Oxidoreductase activity and lipid metabolic process were two of the most significantly enriched GO terms. Figure 2D,E show the 20 most up-enriched and down-enriched pathways identified in the fruit fungus (AnMvsAn), respectively. The peroxisome pathway was significantly up-regulated (Figure 2C), while ribosome biogenesis in eukaryotes was significantly down-regulated (Figure 2C). All of the above results implied that *A. niger* and its mutant strain AnM were different in oxidative metabolism.

### 3.4. Exogenous Inoculation of A. Niger and Its Mutant Strain AnM Influences Internal Browning in Pineapple During Storage

As shown above in this work, *A. niger* (An) and its mutant strain AnM were different in oxidative metabolism. As the essence of pineapple IB was oxidative browning, we inoculated pineapple fruit with *A. niger* and its mutant strain AnM to explore the effect of IB. As shown in Figure 3A,B, the IB incidence in An-inoculated fruit was significantly lower than that in the control and AnM-inoculated after 12 d of storage, while the IB incidence in AnM-inoculated fruit was slightly higher than that in the control; however, the significant difference between was not statistically significant. The results indicated that *A. niger* alleviated pineapple IB, while AnM did not have the same effect on alleviating IB.

### 3.5. Exogenous Inoculation of A. Niger and Its Mutant Strain AnM Influences Flavonoid Contents in Pineapple During Storage

To verify the relationship between IB incidence and flavonoid content in pineapple, flavonoids from pineapple stored at 20 °C for 12 days were determinated via UPLC-MS/MS. A total of 91 flavonoids were identified from pineapple pulp tissues (Appendix A). Among the 91 flavonoids identified were 10 significant, including 1 flavanone, 3 flavones, 3 flavonols, 2 flavonoid carbonosides and 1 flavanol (Table 2). Compared with CK and AnM-inoculated fruit, An-inoculated increased 2 flavones, i.e., Hispidulin (pmp000001) and Hispidulin-7-O-Glucoside (Hmgp002189), and 1 flavanone, i.e., Diosmetin (mws0058) (Figure 3C, Table 2). Meanwhile compared with the control, AnM-inoculated increased 1 flavanol, i.e., Epicatechin (pme0460), 1 flavone, i.e., 5,7-Dihydroxy-3′,4′,5′-trimethoxyflavone (mws1474), 1 Flavonoid carbonoside, i.e., Luteolin-8-C-arabinoside (HJAP012) and 4 flavanols, i.e., Epicatechin (pme0460), Hyperin (mws0061), Quercetin-7-O-glucoside (mws1329) and Quercetin 3-O-α-rhamnopyranosyl (1→2)-[α-rhamnopyranosyl (1→6)]-β-glucopyranoside (Zmgp002857) (Figure 3C, Table 2). These results showed that *A. niger* and its mutant strain AnM regulated flavonoid levels in pineapple pulp.

### 3.6. Changes in Profiles of Transcriptome and Flavonoid Biosynthesis in Pineapple in Response to A. Niger and Its Mutant Strain AnM Inoculation

To further verify that the role of *A. niger* and its mutant strain AnM in regulating pineapple IB involves flavonoids, profiles of the transcriptome and flavonoids of An- and AnM-inoculated pineapple were analyzed. The heatmap exhibiting the expression level of DEGs and DEMs was integrated into the flavonoid biosynthesis pathway (Figure 4A). There were three DEGs associated with flavonol synthase, i.e., *FLS* (*LOC109711383*) *DFR* (*LOC109706489*) and *F3′H* (*LOC109720891*), and four DEMs, i.e., Quercetin, Epicatechin, Vitexin and Luteolin-8-C-arabinoside, in the flavonoid biosynthesis pathway (Figure 4A). The relative expression of *DFR* and *F3′H* (*LOC109720891*) was up-regulated in An.12dvsCK.12 and An.12dvsAnM.12d (Figure 4A,B) but down-regulated in AnM.12dvsCK.12d (Figure 4A,B). In comparison with CK and AnM, An down-regulated *FLS*, while AnM down-regulated it compared to CK (Figure 4A,B).These results suggested that *A. niger* and its mutant strain AnM exert opposing effects on regulating flavonoid biosynthesis through the genes *DFR, F3′H* and *FLS* in the flavonoid biosynthesis pathway.

### 3.7. Changes in Endophytic Fungal Community of Pineapple in Response to A. Niger and Its Mutant Strain AnM Inoculation

To verify the effects of *A. niger* and its mutant strain AnM on the endophytic fungal community from pineapple pulp, pineapples stored at 20 °C for 12 d were analyzed via ITS rRNA.

Appendix A shows that the endophytic fungal community of pineapple in different levels (phylum, family, genus and species) formed a cluster separated from CK, An- and AnM-inoculated pineapple after 12 d of storage, suggesting that *A. niger* and AnM had an impact on the endophytic fungal community.

Figure 5A shows the number of corporate and specific OUTs in pineapple. Following 12 d of storage, there were 295 corporate OUTs in CK, An- and AnM-inoculated pineapple, and the number of specific OUTs was different (Figure 5A), suggesting that *A. niger* and AnM changed the endophytic fungal composition of pineapple. The number of specific OUTs increased sequentially in AnM.12d, CK.12d and An.12d (Figure 5A), suggesting that *A. niger* decreased the absolute abundance of endophytic fungi in pineapple, while AnM increased. Compared to CK, weighted unifrac analysis revealed that *A. niger* drove a significant down-regulation of *beta* diversity in the endophytic fungal composition, whereas AnM administration reduced the alterations (Figure 5B), suggesting that *A. niger* might reduce a certain endophytic fungal composition in pineapple to suppress IB.

At the top-35-species level, *A. niger* treatment increased the relative abundances of seven endophytic fungi (including *Rhodotorula mucilaginosa*, *Kwoniella heveanensis*, *Filobasidium magnum*, *Naganishia albida*, *Cystobasidium* sp., *Naganishia liquefaciens*, *Lysurus cruciatus*) compared with AnM treatment and control and aslo decreased the abundances of nine endophytic fungi (*Bipolaris* sp., *Penicillium oxalicum*, *Phoma herbarum*, *Trichoderma harzianum*, *Fusarium* sp., *Talaromyces* sp., *Hyweljonesia* sp., *Meyerozyma caribbica*, *Russula* sp.) compared with that of AnM and CK (Figure 5C). Meanwhile, the relative abundances of *A. niger* and *Aureobasidium melanogenum* in An-inoculated fruit were increased relative to that of the control (Figure 5C) but decreased relative to that of AnM (Figure 5C). These results indicate that A. niger and its mutant strain AnM changed the endophytic fungi composition in pineapple pulp and that they have an opposite effect on certain endophytic fungi; seven endophytic fungi (including *R. mucilaginosa*, *K. heveanensis*, *F. magnum*, *N. albida*, *N. liquefaciens*, *Cystobasidium* sp., *L. cruciatus*) might be the effective endophytic fungal community.

### 3.8. Correlation Analysis of Flavonoids and Endophytic Fungi in Pineapple

We further performed correlation analyses to investigate the mechanism of changes between the significant 10 flavonoids in content and top 35 endophytic fungi at the species level in pineapple fruit after inoculation with *A. niger* and its mutant strain AnM. Figure 6 shows the correlation analysis between differential flavonoids and the top 35 endophytic fungi at the species level. The flavonoid carbonoside, i.e., Vitexin (mws0048) and Flavanol, i.e., Epicatechin (pme0460), was negatively correlated with six endophytic fungi, including *F. magnum*, *N. albida*, *A. niger*, *A. melanogenum*, *K. heveanensis* and *L. cruciatus*, in An.12dvsCK.12d (Figure 6A), while the flavanol (Epicatechin, pme0460) was positively correlated with two endophytic fungi, including *Cystobasidium* sp. and *R. mucilaginosa*, in An.12dvsAnM.12d and AnM.12dvsCK.12d (Figure 6B,C). Flavones, i.e., Hispidulin (pmp000001) and Hispidulin-7-O-Glucoside (Hmgp002189), and the flavanone, i.e., Diosmetin (mws0058), were positively correlated with six endophytic fungi, i.e., *F. magnum*, *N. albida*, *A. niger*, *A. melanogenum*, *K. heveanensis* and *L. cruciatus*, in An.12dvsCK.12d (Figure 6A), while these three metabolites were not correlated with endophytic fungi in AnM.12dvsCK.12d (Figure 6C). All of the above results show that An-inoculated pineapple had a positive significant correlation between the contents of three flavonoids (Hispidulin, Hispidulin-7-O-Glucoside, Diosmetin) and the relative abundance of six endophytic fungi (*F. magnum*, *N. albida*, *A. niger*, *A. melanogenum*, *K, heveanensis*, *L. cruciatus*), while two flavonoids (Vitexin, Epicatechin) had a significant negative correlation between those six endophytic fungi in An-inoculated pineapple.

## 4. Discussions

Pineapple is a subtropical and tropical fruit favored by customers across the globe; however, its international trade is severely restricted by its short-term shelf-life due to internal browning (IB) at postharvest. IB in pineapples, a physiological disorder, has been previously studied [7]. The influence of *A. niger* on plant resistance is well-documented [79,80]. Endophytes have been used to manage infectious diseases in plants [81,82,83,84]. Our previous research demonstrated, for the first time, that *Penicillium* sp., an endophytic fungus isolated from pineapple, can effectively control IB in postharvest pineapples [17]. In this study, the inoculation of the endophytic fungus *A. niger* (An) of pineapple fruit could effectively reduce IB symptom and incidence (Figure 3A,B), suggesting that *A. niger* suppressed the development of IB of postharvest pineapple. The findings indicated again that endophytic fungi not only served as antagonistic agents against pathogenic infections in plants but also had the potential to control physiological disorders in plants (non-infectious diseases in plants). Given that IB of pineapple is a physiological disorder [7], the results of this study may suggest a regulatory role of endophytic fungi in the plant physiological status. How plant endophytes regulate the plant physiological status is a subject worthy of further in-depth investigation. To investigate this problem, we obtained and screened a mutant strain of *A. niger* (marked AnM) via mutagenesis breeding. Following a 12 d storage, IB incidence in the AnM-inoculated fruit was not significantly different from that of CK (the control) but was significantly higher than that of the An-inoculated (Figure 3A,B), suggesting that AnM could not control IB of pineapple. The transcriptome data of the fungus showed that there were 7548 genes that were co-expressed in *A. niger* and its mutant strain AnM, with 368 specific genes of *A. niger,* whereas the mutant strain AnM expressed 1640 specific genes (Figure 2A). A total of 2391 DEGs were discovered in AnMvsAn (the endophytic fungus *A. niger* compared to the mutant strain AnM), including 1279 up-regulated DEGs and 1112 down-regulated (Figure 2B). These results suggested that *A. niger* and its mutant strain AnM have dramatically different patterns at the transcriptional level under natural conditions, and there could still be something in common between them, and AnM could not control IB of pineapple. The results mentioned above imply that the mutant strain could serve as a negative control strain for studying the mechanism by which *A. niger* controls IB in pineapples.

Previous studies have proved that DFR, F3′H and FLS are the key enzymes in the flavonoid biosynthesis pathway [85,86,87]. Additionally, the expression levels of *DFR*, *F3′H* and *FLS* genes were positively correlated with the flavonoid contents in plants [86,88]. In this work, compared to CK and AnM, An (*A. niger*) up-regulated some key DEGs involved in flavonoid biosynthesis pathways, including *DFR* (*LOC109706489*) and *F3′H* (*LOC109720891*), and down-regulated *FLS* (*LOC109711383*) following 12 d of storage, while AnM down-regulated *DFR* and *F3′H* and up-regulated *FLS* in comparison with CK (Figure 4A,B). The metabolome results showed that *A. niger* increased the contents of three flavonoids in pineapple and decreased the contents of seven flavonoids when compared to CK (the control) and AnM after 12 d of storage (Figure 3C, Table 2). This result suggested that the alleviation of IB in pineapple by *A. niger* was associated with flavonoid biosynthesis, which involved the up-regulation of the key genes *DFR* and *F3′H*, along with the down-regulation of *FLS* in the flavonoid biosynthesis pathway.

The essence of IB is enzymatic oxidation. Flavonoids serve as natural antioxidants [89] and can help control plant diseases [18,19,20,90]. The antioxidant capacity of pineapple is an ‘early’ response to IB [91], and the enhancement of this capacity is closely related to the inhibition of IB in pineapples [9]. Flavonoids possess antioxidant properties [29,65,89,92]. For instance, flavonoids and their glycosides from Inula britannica showed profound antioxidant activity [93]. Studies have also shown that the antioxidant capacity of plants was positively correlated with their flavonoid content [30,66,94,95,96]. As reported by Zhang et al. [96], lactic acid bacteria strains with high antioxidant activity were employed for alfalfa silage, which improved the antioxidant status of ensiled alfalfa and increased the level of total flavonoids while decreasing the losses of α-tocopherol and β-carotene. The fungal endophyte MD89, isolated from pigeon pea, secretes apigenin (a kind of flavonoid) that could enhance the antioxidant activity of HepG2 cells by increasing the activities of superoxide dismutase and glutathione reductase [97]. However, to the authors’ knowledge, no research has investigated whether it is possible to regulate IB undergoing flavonoid synthesis in harvested pineapple. In this study, among the 20 flavonoids detected, compared to ST.0d (storage at 0 d), 19 of them in ST.12d (after 12 d storage) in pineapple fruit were decreased, with the exception of Luteolin-8-C-arabinoside (Figure 1 C). ST.0d was the pineapple fruit just harvested that showed no IB, while ST.12d was the fruit exhibiting serious IB at the storage of 12 d (Figure 1A,B). These results proved that the biosynthesis and degradation of flavonoids occur simultaneously with the aggravation of IB during pineapple storage; the decrease in flavonoid content in pineapple may be one of the contributing factors to the aggravation of IB in pineapple. Romsdahl et al. [80] reported the antioxidative effects of *A. niger* strains against ultraviolet radiation. The transcriptomic analysis of fungus in this study implied that *A. niger* and its mutant strain AnM exhibited differences in oxidative metabolism (Figure 2C–E). In addition, here, we show *A. niger*-inoculated significantly reduced IB incidence compared with CK following 12 d of storage, while the IB incidence of AnM-inoculated was not different from that of CK. This might imply that *A. niger* alleviated IB of pineapple related to its own antioxidant system. We also found that the contents of three flavonoids (i.e., Hispidulin, Diosmetin, Hispidulin-7-O-Glucoside) in An.12d were significantly higher than those in CK.12d and AnM.12d, while there was no difference in the contents of these three flavonoids in AnM.12dvsCK.12d (Figure 3C, Table 2), and they were significantly low in ST.12dvsST.0d (Figure 1C). This suggested that *A. niger* could improve the antioxidant capacity by increasing the content of certain flavonoids (Hispidulin, Diosmetin, Hispidulin-7-O-Glucoside) in pineapple, while AnM could not improve the antioxidant capacity of pineapple, as it did not increase the contents of these flavonoids. All the above results concluded that *A. niger* can enhance the antioxidant capacity of pineapple by increasing the contents of certain flavonoids (Hispidulin, Hispidulin-7-O-Glucoside and Diosmetin) in pineapple fruit and inhibit the occurrence of IB. This result was consistent with previous studies showing that the endophytic fungi *Bipolaris* sp. and *Phoma* sp. enhanced the antioxidant stress capacity of crops by increasing the contents of flavonoids [98]. In addition, the results showed that after *A*. *niger* treatment, the activity of PPO (an enzyme that degrades flavonoids) in pineapple was significantly lower than that of the control, and the activity of POD (an antioxidant defense enzyme) was significantly increased, which was also consistent with the results of this study. However, it still needs to be further studied whether these three flavonoids are produced by *A. niger* and then released into pineapple fruit to control IB or if the fruit produces them in response to induction by *A. niger*.

The inoculation of endophytes with plants can control diseases of plants by modifying the fungal and bacterial community composition and diversity on the plant [68,69]. Ganley et al. [68] showed that fungal endophytes from *Pinus monticola* were efficient at improving survival in the *P. monticola* plant against white pine blister and regulating the structure of microbial communities in the host plant. The Asian citrus psyllid (ACP) is a notorious Rutaceae plant pest that results in Huanglongbing (HLB) [99] and sooty mold [100] of citrus. It has been reported that the positive linear correlations were discovered between the resistance level of PCA and mean relative abundances of two endophytes (*A. niger* and *Aureobasidium pullulans*) in the Asian citrus [67]. In this study, we obtained pineapple pulp samples under sterile conditions and used these samples for endophytic fungal community analysis via ITS rRNA amplicon sequencing, ensuring that our subsequent analysis was based on endophytic fungal analysis. *A. niger* reduced the number of unique endophytic fungi and the *beta* diversity of the endophytic fungal community in pineapple, while AnM was opposite (Figure 5A,B), and the endophytic fungal community of pineapple at the phylum, family, genus and species levels formed a cluster separated from CK, An- and AnM-inoculated pineapple after 12 d of storage (Appendix A), suggesting that *A. niger* regulated the composition, abundance and diversity of the endophytic fungal community in postharvest pineapple. At the species level, *A. niger* inoculation increased the relative abundances of seven endophytic fungi (including *R. mucilaginosa*, *K. heveanensis*, *F. magnum*, *N. albida*, *Cystobasidium* sp., *N. liquefaciens*, *L. cruciatus*) compared with AnM treatment and the control, which also decreased the abundances of nine endophytic fungi (*Bipolaris* sp., *P. oxalicum*, *P. herbarum*, *T. harzianum*, *Fusarium* sp., *Talaromyces* sp., *Hyweljonesia* sp., *M. caribbica*, *Russula* sp.) compared with that of AnM and the control (Figure 5C). These results indicated that *A. niger* and its mutant strain AnM changed the endophytic fungal abundance in pineapple; these endophyte fungi (*R. mucilaginosa*, *K. heveanensis*, *F. magnum*, *N. albida*, *Cystobasidium* sp., *N. liquefaciens*, *L. cruciatus*) may be beneficial to control pineapple IB. Meanwhile, the relative abundances of *A. niger* and *A. melanogenum* in An-inoculated fruit were increased relative to that of the control (Figure 5C) but decreased relative to that of AnM (Figure 5C), implying that increasing the abundance of *A. niger* and *A. melanogenum* to a certain extent is conducive to controlling IB in pineapples; there may be a threshold of the abundance of *A. niger* and *A. melanogenum* in pineapple that controls IB; however, this threshold needs further study. All the above results concluded that *A. niger* reduces IB by regulating the community structure and population abundance of endophytic fungi in pineapple fruit.

In addition, the combined metabolomic and microbiome analysis showed that An-inoculated pineapple had a significant and positive correlated between the content of three flavonoids, i.e., Hispidulin-7-O-Glucoside (Hmgp002189), Hispidulin (pmp000001) and Diosmetin (mws0058), and the relative abundances of six endophytic fungi, i.e., *F. magnum*, *N. albida*, *A. niger*, *A. melanogenum*, *K. heveanensis* and *L. cruciatus,* compared with that of CK following 12 d of storage (Figure 6A). However, no such result was found in AnM.12dvsCK.12d (Figure 6C). Meanwhile, An alleviated IB, whereas AnM was unable to control IB (Figure 3A,B). An increased the contents of these three flavonoids (Figure 3C, Table 2) and the relative abundance of these six endophytic fungi (Figure 5C) compared with the CK. In comparison with AnM, An increased the contents of these three flavonoids (Figure 3C, Table 2) and increased the relative abundances of four of these endophytic fungi, including *F. magnum*, *N. albida*, *K. heveanensis* and *L. cruciatus* (Figure 5C). All of these results indicated firstly that the exogenous *A. niger* positively regulated the interaction between the level of these three flavonoids (Hispidulin-7-O-Glucoside, Hispidulin and Diosmetin) and the relative abundance of these six beneficial endophytic fungi (*F. magnum*, *N. albida*, *A. niger*, *A. melanogenum*, *K. heveanensis* and *L. cruciatus*) in pineapple fruit to control IB. Ochratoxin and fumonisin were mycotoxins resulting in a serious health hazard and produced by the genera *Aspergillus*, *Penicillium* and *Fusarium* [101,102]. In this study, some species of endophytic fungi of *Aspergillus, Penicillium* and *Fusarium* could be detected and were species among the top 35 in relative abundance at the species level in An.12d fruit (Figure 5C). Our previous results suggested that no ochratoxin and fumonisin was detected in the *An*-inoculated pineapple fruit, and even the genes of ochratoxin and fumonisin synthesis were not expressed, indicating that the strain of *A. niger* strain did not produce ochratoxin and fumonisin and could ensure the food safety of pineapple after the inoculation of *A. niger* [103].

## 5. Conclusions

Our study indicated that the biosynthesis and degradation of flavonoids occurred simultaneously with the aggravation of pineapple IB during pineapple storage; the aggravation of pineapple IB was closely related to a decrease in flavonoid contents in pineapple. The application of the endophyte *A. niger* from pineapple reduced IB, involving the regulation of DEGs of flavonoid biosynthesis and the endophyte fungal community structure and population abundance in pineapple fruit. Both increasing the contents of three flavonoids (Hispidulin, Hispidulin-7-O-Glucoside, Diosmetin) and increasing the relative abundances of six endophytic fungi (*F. magnum*, *N. albida*, *A. niger*, *A. melanogenum*, *K. heveanensis*, *L. cruciatus*) were beneficial to control IB of pineapple. The levels of the three flavonoids and six endophytic fungi mentioned above had a significant positively correlation. In conclusion, *A. niger* alleviated IB mainly by enhancing the contents of flavonoids and the relative abundances of endophytic fungi and regulating the interaction between the flavonoid content and endophytic fungi abundance in pineapple (Figure 7).

## Figures and Tables

**Figure 1 jof-10-00794-f001:**
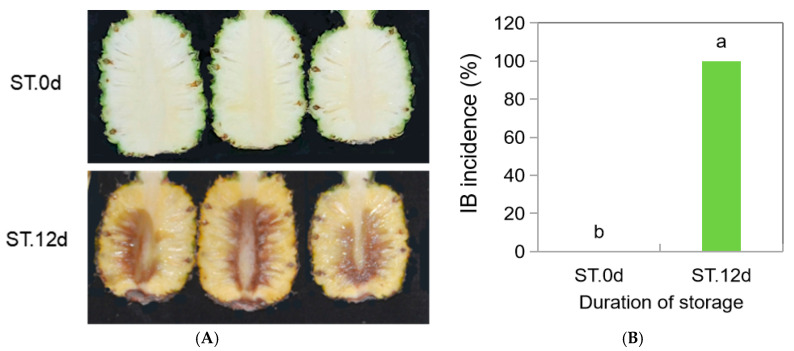
The changes in internal browning (IB) symptoms (**A**), IB incidence (**B**) and the flavonoid contents (**C**) of pineapple fruit following 12 d of storage at ambient temperature. St.0d, before storage; St.12d, storage at 12 days. Values are the means ± SE (*n* = 3). (**B**) The different letters are significantly different (*p* ≤ 0.05). (**C**) The numbers show Log2fold-change; St.12dvsSt.0d represents pineapple storage at 12 days (St.12d) compared to before storage (St.0d); * show the presence of isomers of the substance.

**Figure 2 jof-10-00794-f002:**
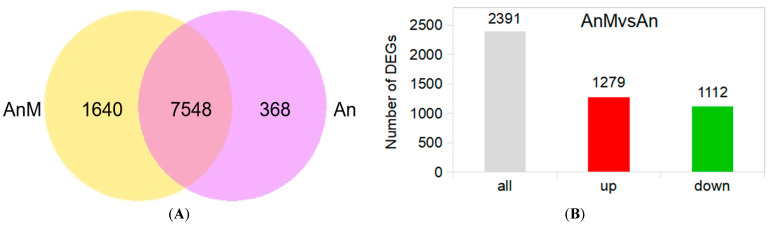
Analysis of transcriptome data of endophyte *A. niger* and its mutant strain AnM. (**A**) Venn diagram of gene expression. (**B**) Histogram of differential expression gene (DEG) number. (**C**) Histogram of GO term enrichment of DEGs; *n*, the number of DEGs; *, significant enrichment GO term; BP (in red font), Biological Proces; CC (in green font), Biological Proces; FM (in blue font), Molecular Function. (**D**) Histogram of KEGG enrichment of up-regulated DEGs; N_up_, the number of up-regulated DEGs; * and red font, significant enrichment KEGG pathway. (**E**) Histogram of KEGG enrichment pathway of down-regulated DEGs; N_down_, the number of down-regulated DEGs; * and red font, significant enrichment KEGG pathway. An, endophyte *A. niger*; AnM, the mutant strain of *A. niger* marked AnM.

**Figure 3 jof-10-00794-f003:**
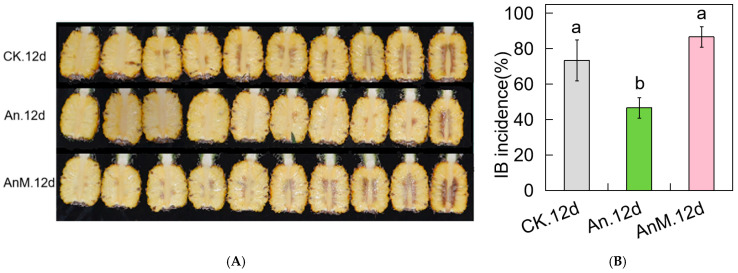
The changes in IB and flavonoid content of pineapple responsible for *A. niger*- and AnM-inoculated following 12 d of storage at ambient temperature. (**A**) IB symptom. (**B**) IB incidence; the different letters are significantly different (*p* ≤ 0.05). (**C**) The content of flavonoids in pineapple. CK.12d, the control fruit following 12 d of storage; An.12d, *A. niger*-inoculated fruit following 12 d of storage; AnM.12d, the mutant strain AnM-inoculated fruit following 12 d of storage; * show the presence of isomers of the substance.

**Figure 4 jof-10-00794-f004:**
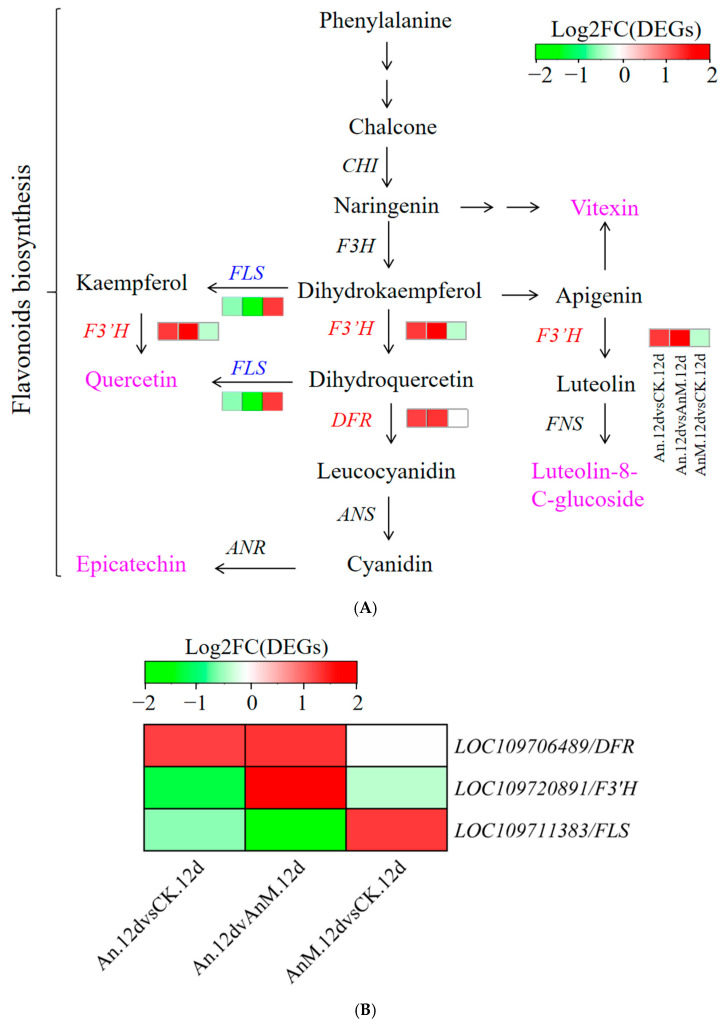
Effects of *A. niger* and AnM on profiles of transcriptome and flavonoids in pineapple pulp following 12 d of storage at ambient temperature. (**A**) Reconstruction of the flavonoid biosynthesis pathway with the DEGs. (**B**) Differential expression of genes in flavonoid biosynthesis pathway of pineapple responsible for *A. niger* and AnM (date of transcriptome). DEGs, differentially expressed genes. FC, fold-change. CK.12d, the control fruit following 12 d of storage; An.12d, *A. niger*-inoculated fruit following 12 d of storage; AnM.12d, the mutant strain AnM-inoculated fruit following 12 d of storage. CHI, chalcone isomerase; F3H, flavanone 3-hydroxylase; F3′H, flavonoid 3′-monooxygenase; DFR, dihydroflavonol 4-reductase; ANS, anthocyanin synthase; FLS, flavonol synthase; ANR, anthocyanidin reductase. The charts in green and red of Figure A indicate the down- and up-regulated genes or flavonoids, respectively. The red and blue fonts indicate DEGs that are significantly up-regulated and down-regulated in the flavonoid biosynthesis pathway for An.12d vs CK.12d and An.12d vs AnM.12d. The blue fonts indicate the DEMs (differentially metabolites) in flavonoid biosynthesis pathway for An.12d vs CK.12d and An.12d vs AnM.12d.

**Figure 5 jof-10-00794-f005:**
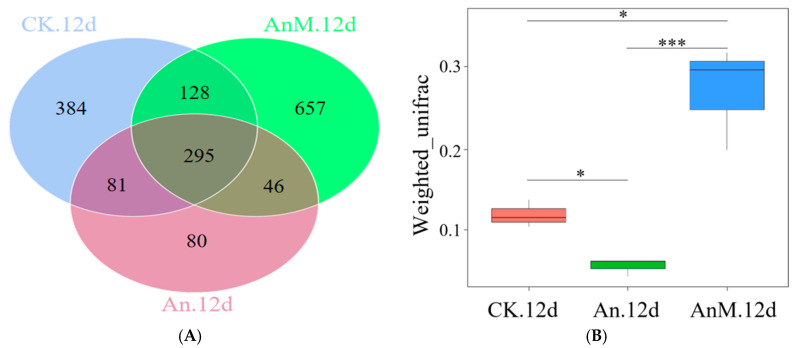
The changes in endophytic fungal communities of pineapple fruit responsible for *A. niger*- and AnM-inoculated following 12 d of storage at ambient temperature (date from ITS rRNA amplicon sequencing). (**A**) The Venn diagram of endophyte fungi in pineapple based on OTU. (**B**) The *beta* diversity of endophytic fungi in pineapple based on the weighted unifrac analysis (* represents *p* < 0.05; *** represents *p* ≤ 0.001). (**C**) The relative abundance of the top 35 endophytic fungi at the species level in pineapple. CK.12d, the control fruit following 12 d of storage; An.12d, *A. niger*-inoculated fruit following 12 d of storage; AnM.12d, the mutant strain AnM-inoculated fruit following 12 d of storage.

**Figure 6 jof-10-00794-f006:**
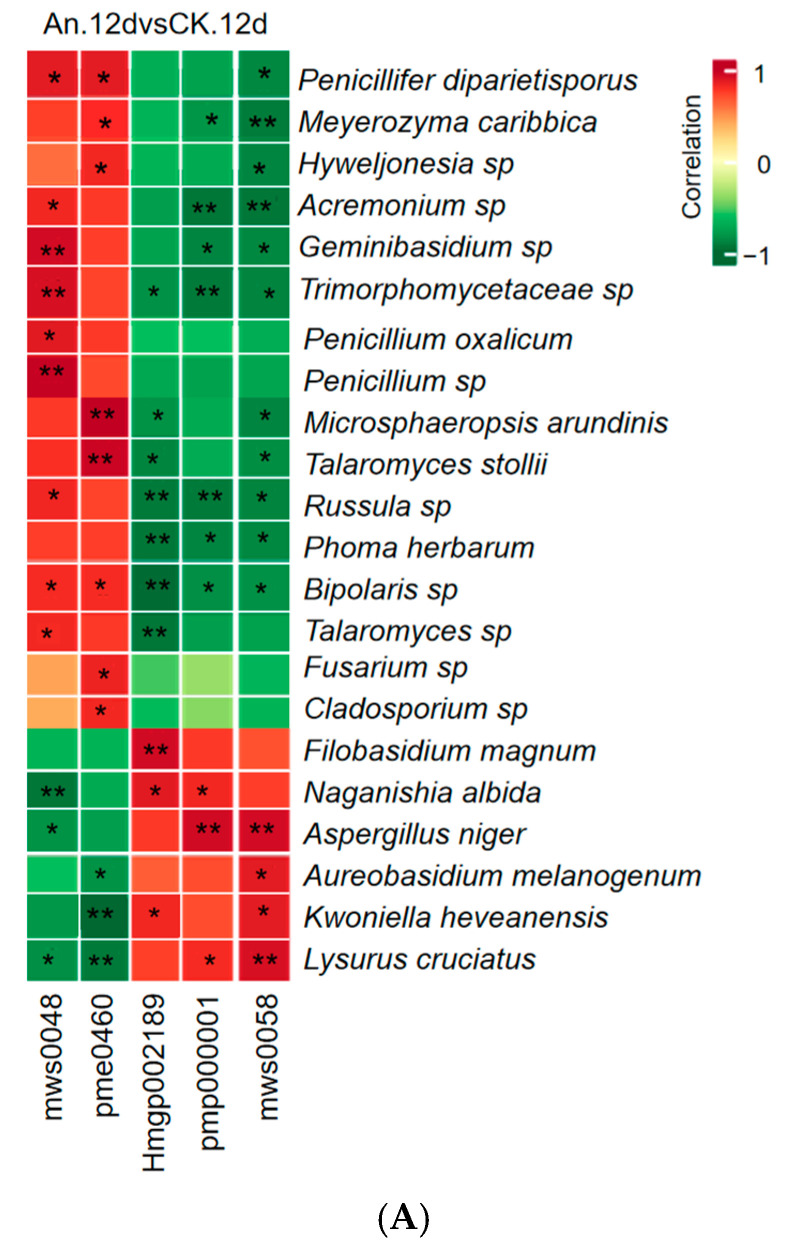
Association map of the two-tiered analyses integrating endophytic fungi and flavonoids in pineapple pulp responsible for *A. niger*- and AnM-inoculated following 12 d of storage at ambient temperature. (**A**) An.12dvsCK.12d. (**B**) An.12dvsAnM.12d. (**C**) AnM.12dvsCK.12d. mws0048, Vitexin (Apigenin-8-C-Glucoside); pmp000001, Hispidulin; Hmgp002189, Hispidulin-7-O-Glucoside; pmp000001, Hispidulin (5,7,4′-Trihydroxy-6-methoxyflavone)*; mws0058, Diosmetin (5,7,3′-Trihydroxy-4′-methoxyflavone)*. The intensity of the colors indicates the degree of association (red, significant positive correlation; green, significant negative correlation). *p*-value < 0.05 of test of correlation coefficient is a significant difference; *, *p* < 0.01; **, *p* < 0.001.

**Figure 7 jof-10-00794-f007:**
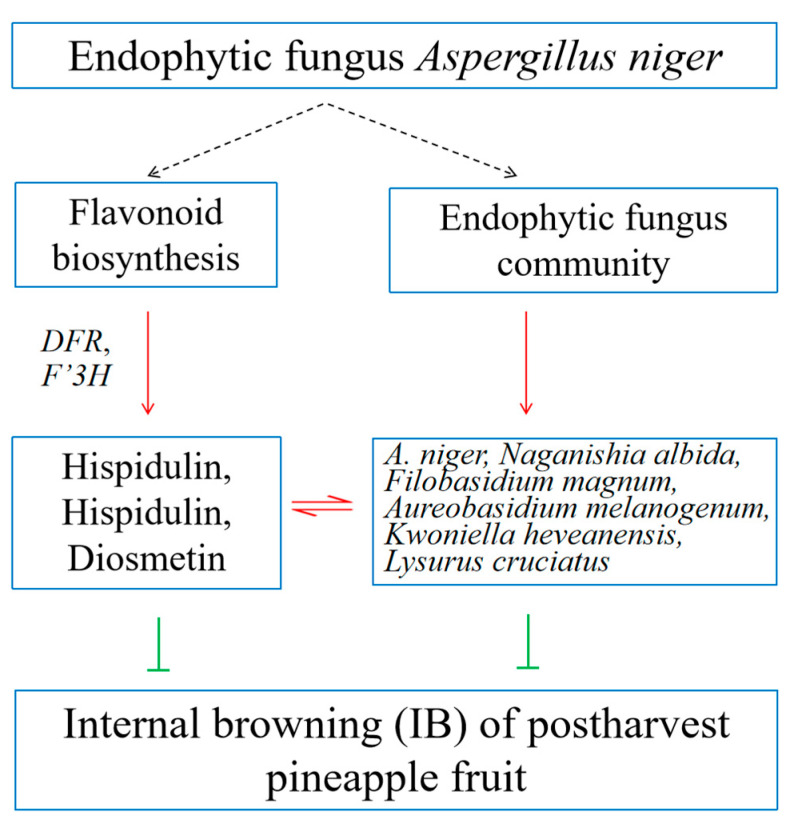
Simplified hypothetical model for the mechanism of pineapple internal browning suppression regulated by *A. niger*. The black arrow ‘
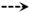
’ represents trends in regulation, the red arrow ‘→’ represents trends within an increase or up-regulation, the red mark ‘⇌’ represents trends within positive-regulation and the green mark ‘┤’ represents the trend within suppression.

**Table 1 jof-10-00794-t001:** Overview of *A. niger* and its mutant strain AnM transcriptome sequencing and assembly.

Sample	An_1	An_2	An_3	AnM_1	AnM_2	AnM_3
Raw reads	21,626,922	20,851,543	23,667,532	20,562,749	20,182,438	21,688,258
Raw bases	6.49 Gb	6.26 Gb	7.1 Gb	6.17 Gb	6.05 Gb	6.51 Gb
Clean reads	42,071,514	40,611,978	46,066,832	40,151,386	39,302,936	42,323,184
Clean bases	6.31 Gb	6.09 Gb	6.91 Gb	6.02 Gb	5.9 Gb	6.35 Gb
Error rate	0.03	0.03	0.03	0.03	0.03	0.03
Q20 percentage	97.62%	97.68	97.64	97.94	97.69	97.58
Q30 percentage	93.57%	93.67	93.59	94.17	93.69	93.42
GC percentage	54.8%	54.91	54.99	54.61	54.24	54.28
Total mapped	21,035,757	20,305,989	23,033,416	20,075,693	19,651,468	21,161,592
Mapping percent	89.71%	90.60%	90.62%	89.98%	88.55%	89.02%

Note: An, the endophytic fungus *A. niger*; AnM, the mutant strain of *A. niger* marked AnM.

**Table 2 jof-10-00794-t002:** Exogenous inoculation *A. niger* and its mutant strain AnM influences internal browning and flavonoid contents in pineapple during storage.

Class	Compounds	Index	CAS	Molecular Weight (Da)	CK.12d	An.12d	AnM.12d
Flavone	Hispidulin (5,7,4′-Trihydroxy-6-methoxyflavone) *	pmp000001	1447-88-7	300.06	3.61 × 10^3^ b	7.41 × 10^3^ a	3.43 × 10^3^ b
	Hispidulin-7-O-Glucoside	Hmgp002189	17680-84-1	462.12	3.53 × 10^4^ b	9.85 × 10^4^ a	3.85 × 10^4^ b
	5,7-Dihydroxy-3′,4′,5′-trimethoxyflavone	mws1474	18103-42-9	344.09	2.02 × 10^3^ b	2.82 × 10^3^ b	6.56 × 10^3^ a
Flavanone	Diosmetin (5,7,3′-Trihydroxy-4′-methoxyflavone) *	mws0058	520-34-3	300.06	9.00 b	8.31 × 10^3^ a	9.00 b
Flavanol	Epicatechin *	pme0460	490-46-0	290.08	1.60 × 10^5^ b	6.91 × 10^4^ c	4.90 × 10^5^ a
	Quercetin-3-O-galactoside (Hyperin) *	mws0061	482-36-0	464.10	4.87 × 10^4^ b	3.42 × 10^4^ b	7.06 × 10^4^ a
	Quercetin-7-O-glucoside *	mws1329	491-50-9	464.10	2.54 × 10^4^ b	1.38 × 10^4^ b	3.17 × 10^4^ a
	Quercetin 3-O-α-rhamnopyranosyl (1→2)-[α-rhamno-pyranosyl (1→6)]-β-glucopyranoside	Zmgp002857	-	756.21	1.35 × 10^4^ b	1.09 × 10^4^ b	2.29 × 10^4^ a
Flavonoid carbonoside	Vitexin (Apigenin-8-C-Glucoside) *	mws0048	3681-93-4	432.11	1.15 × 10^4^ a	4.61 × 10^3^ b	8.49 × 10^3^ b
	Luteolin-8-C-arabinoside	HJAP012	-	418.09	3.38 × 10^4^ b	2.68 × 10^4^ b	5.54 × 10^4^ a

Note: CK.12d, the control fruit following 12 d storage; An.12d, *A. niger*-inoculated fruit following 12 d storage; AnM.12d, the mutant strain AnM-inoculated fruit following 12 d storage; *, the presence of isomers of the substance; the lowercase letters (a, b, c), significant differences between flavonoid contents in pineapple under different treatments (*p* < 0.05); red font and blue font, flavonoid content in the treatment is significant up-regulated and significant down-regulated, respectively.

## Data Availability

The original contributions presented in the study are included in the article/Appendix A, further inquiries can be directed to the corresponding author.

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
