# Peer review of "The Role of Aspergillus niger in Regulating Internal Browning Involves Flavonoid Biosynthesis and the Endophytic Fungal Community of Pineapple"

_jof, 2024, doi:10.3390/jof10110794_

Round 1

Reviewer 1 Report

Review of Mansucript 3190272

“The role of Aspergillus niger in regulating internal browning involves flavonoid biosynthesis and endophytic fungal community of pineapple”

The manuscript provides a description of the effect of Aspergillus niger in regulating the internal browning in which the flavonoids biosynthesis and endophytic fungal community of pineapple. Some issues are pointed out in the next.

-please describe briefly the source of the endophytic Aspergillus niger.

-The authors described that the pulp of pineapple fruit was used to determine flavonoids and total RNA. Did the authors also used the pulp for the analysis of endophytic fungal community? This is because Aspergillus niger (both native and mutant) was spray inoculated. This means that the treatments were for the entire fruit.

-The storage of 12-d of pineapple corresponds to a rule or how is the establishment for 12 days of storage.

- The spore suspension was made in H2O sterile or in a regulated solution.

-The authors mentioned that 7548 genes were co-expressed of the A. niger and its mutant. Next, the  authors mentioned 368 specific genes of native A. niger, whereas the mutant AnM expressed 1640 specific genes. But what mean these results means. There is no discussion regarding such results. How the mutant can express more specific genes than the native.  

No comments

Reviewer 2 Report

In my opinion, the main concern I have with this manuscript is that the authors do not account for fungi that may reside on the fruit surface after harvest, treating them all as endophytic fungi. As a result, the conclusions drawn may not be fully supported.

How can the endophytic fungal community in a harvested fruit change after it has already been cut from the plant?

Additionally, the selection process for the Aniger mutant is unclear. Did the authors obtain other mutants and select the one presented, or was this the only mutant identified?

-

Reviewer 3 Report

This manuscript presents some interesting results suggesting the strain of Aspergillus niger used can reduce internal browning in pineapple that is associated with increases in some flavonoids and facultative endophytic microorganisms. For the most part methodology appears to be appropriate, but some clarification is needed in some locations. Interpretation of results may need some revision upon further reflection. Of most concern is the authors did not mention or evaluate the strain of Aspergillus niger used for ochratoxin (a known nephrotoxin and renal carcinogen) production, which is reported for several strains of this fungus.  The authors appear to have the expertise to make this determination, preferably in the treated pineapple fruit as well as with cultures.

Flavonoids have differing antioxidant capabilities, so it would be appropriate to include a table with the isolated flavonoids indicating their relative antioxidant capabilities (information which is available in published literature) and then relate that information to the ones the authors think are reducing the browning, on a concentration-based determination (i.e. are levels high enough to inhibit oxidation).  Maybe the ones that are decreased are the substrates for the oxidation reaction, so their decrease is contributing to the reduction in browning.

It would also be useful to discuss any reported potential negative impacts of the endophytes isolated (such as toxin production) and include any information on their previously reported ability to inhibit browning or stimulate flavonoid production in other crops or plants. Some species of Fusarium and Penicillium can produce toxins and carcinogens, and an endophytic strain of Penicillium oxalicum is a reported ochratoxin producer.

There seem to be some discrepancy between what is shown in the Figures and what is stated in the text, notably for Figures 3 and 6, so it is difficult to currently interpret relevance.

Authors should specify in the text the expression of relevant pineapple oxidizing enzymes that may be using flavonoids as substrates to cause the browning, such as polyphenol oxidase and peroxidase. Many other oxidizing enzymes would be irrelevant.

The Results section numbers go from 3.5 to 3.7. Is there a section 3.6 that is missing?

Is there any information in the literature on the effect of the other endophyte species on plant flavonoid production or ability to inhibit activity of oxidative enzymes, whether by production of such things as secondary metabolites or proteases? If so, that information should be added to the Discussion.

There were several typos noted throughout the manuscript (e.g. 346, 416, 456, 520, 548, 733, 817), so the authors should recheck the paper. The English also need some polishing as there were several examples of improper word and tense use (e.g. lines 94, 157, 171, 197, 257, 400, 412, 421, 444, 512, 561) and failure to italicize scientific names (e.g. 413-426, 703, 853).

Round 2

Reviewer 3 Report

The authors have done a nice job with the revisions. Only a few minor issues remain.

It would be valuable to provide citations indicating the flavonoids (hispidulin, and its glucoside, diosmetin) that are increased by An treatments can act as antioxidants. Possible citations that may be useful themselves or have more relevant information cited are Molecules 28: 7910 and Journal of Natural Products 36: 1-4.

While it is understandable that the authors would be reluctant to specifically mention that An affected PPO and POD activity if they are the subject of an unpublished manuscript, it would be appropriate to do so, at least mentioning as unpublished data, to better tell the full story.

Author Response

Comments 1: It would be valuable to provide citations indicating the flavonoids (hispidulin, and its glucoside, diosmetin) that are increased by An treatments can act as antioxidants. Possible citations that may be useful themselves or have more relevant information cited are Molecules 28: 7910 and Journal of Natural Products 36: 1-4.

Response 1: Thank you for your valuable suggestion. We have citated this two articles in the revised manuscrip (Line 545 and Line 548-549).

Comments 2: While it is understandable that the authors would be reluctant to specifically mention that An affected PPO and POD activity if they are the subject of an unpublished manuscript, it would be appropriate to do so, at least mentioning as unpublished data, to better tell the full story.

Response 2: Your concern is justified. In fact, we found that An decreased the activity of PPO and increased the activity of POD in pineapple fruit (unpublished data), so as to maintain the high flavonoids content in pineapple and improve the antioxidant property of pineapple. We also have mentioned these date in the revised manuscrip (Line 585-589).